# Time line of redox events in aging postmitotic cells

**Nicolas Brandes[1][†], Heather Tienson[1][†], Antje Lindemann[1], Victor Vitvitsky[2], Dana Reichmann[1][‡], Ruma Banerjee[2], Ursula Jakob[1,2]***

[1]Department of Molecular, Cellular, and Developmental Biology, University of Michigan, Ann Arbor, United States; [2]Department of Biological Chemistry, University of Michigan Medical School, Ann Arbor, United States

**Abstract** The precise roles that oxidants play in lifespan and aging are still unknown. Here, we report the discovery that chronologically aging yeast cells undergo a sudden redox collapse, which affects over 80% of identified thiol-containing proteins. We present evidence that this redox collapse is not triggered by an increase in endogenous oxidants as would have been postulated by the free radical theory of aging. Instead it appears to be instigated by a substantial drop in cellular NADPH, which normally provides the electron source for maintaining cellular redox homeostasis. This decrease in NADPH levels occurs very early during lifespan and sets into motion a cascade that is predicted to down-regulate most cellular processes. Caloric restriction, a near-universal lifespan extending measure, increases NADPH levels and delays each facet of the cascade. Our studies reveal a time line of events leading up to the system-wide oxidation of the proteome days before cell death.

***For correspondence:** ujakob@umich.edu

[†]The first two authors contributed equally to this work

[‡]**Present address:** Department of Biological Chemistry, Hebrew University of Jerusalem, Jerusalem, Israel

**Competing interests:** The authors have declared that no competing interests exist

**Reviewing editor**: Karsten Weis, University of California-Berkeley, United States

## Introduction

Most living animals undergo a physiological decline with age. Yet, despite decades of intense study, no consensus has emerged regarding the primary cause of this decline. One leading hypothesis is the free radical theory of aging, which postulates that aging is caused by an accumulation of oxidative damage to cellular macromolecules (*Harman, 1956*). Many lines of correlative evidence support this theory (*Muller et al., 2007*). However, while these studies confirm the general notion that oxidative damage is associated with aging, recent studies in mice have generated conflicting results as few of the genetic manipulations targeting conserved antioxidant genes showed the predicted effects on lifespan (*Perez et al., 2009*). Hence, the jury is still out on the question of whether oxidative damage is a cause of aging or simply a consequence (*Salmon et al., 2010*).

One obstacle in defining the role of oxidants in aging is our lack of knowledge of when, or even if, reactive oxygen species (ROS) accumulation causes physiological alterations that are severe enough to affect the lifespan of an organism and whether manipulation of the onset of oxidative stress will alter lifespan. So far, the most commonly used read-out for oxidative protein damage involves detection of protein carbonylation (*Shacter et al., 1994*; *Levine, 2002*). However, neither the extent of carbonylation nor the specific effect(s) of carbonylation on protein activity are easily assessed. To get a better handle on evaluating oxidative protein modifications, we developed a highly sensitive and fully quantitative mass spectrometry-based redox technique (i.e., OxICAT) that allows us to determine the in vivo oxidation status of hundreds of different protein thiols in organisms and to identify the proteins affected (*Leichert et al., 2008*). We recently used OxICAT in *Saccharomyces cerevisiae* to quantify the steady-state oxidation status of almost 400 different yeast protein thiols and identify those proteins that contain peroxide and redox-sensitive cysteines (*Brandes et al., 2011*). We then reasoned that by monitoring the exact oxidation status of these proteins during the chronological

**eLife digest** While most animals experience a physiological decline as they age, the underlying cause of this decline is not fully understood. According to the free radical theory of aging, chemicals known as reactive oxygen species build up in the body and then cause damage to various components within cells, including DNA and proteins. These species, which include hydrogen peroxide and peroxynitrite, can cause substantial oxidative damage. However, while there is definitely a relationship between aging and reactive oxygen species, it remains possible that oxidative damage is a byproduct of aging rather than the cause of it.

In the past researchers have measured the carbonylation of proteins (that is, the oxidation of certain amino acids in proteins) as a proxy for damage caused by reactive oxygen species, but this method has a number of shortcomings. More recently, it has become possible to quantify the oxidation state of cysteine, an amino acid that contains sulfur, in proteins using a technique based on mass spectrometry. Building on previous work in which they used this technique to measure the oxidation state of 300 proteins in vivo in the yeast *Saccharomyces cerevisiae*, Brandes et al. have now determined how the oxidation state of these proteins changes over the lifespan of *S. cerevisiae*, which is a popular model system for analyzing aging in cells that are in a high metabolic state but are no longer dividing. This made it possible to identify protein targets that might—as a result of changes in their oxidation state caused by reactive oxygen species—contribute to the physiological alterations observed in aging organisms. It was also possible to establish a clear connection between the onset and extent of oxidative stress and lifespan.

Brandes et al. discovered that several days before the yeast cells died, they underwent a sudden and global 'redox collapse' in which ~80% of the 300 proteins being studied experienced an increase in their oxidation state (i.e., they lost electrons). This event was preceded by a large drop in the level of NADPH, a coenzyme that, by being a source of electrons, helps to counterbalance the removal of electrons by reactive oxygen species within cells. The drop in the concentration of NADPH occurred very early in the life cycle of the yeast, and set in motion a series of events that down-regulated most cellular processes. Intriguingly, these findings are consistent with the effect of caloric restriction, a condition that is known to extend the lifespan of animals. Caloric restriction increases cellular NADPH and delays the down-regulation of cellular processes.

Brandes et al. propose that the underlying cause of aging is not the accumulation of reactive oxygen species: rather, these results suggest that aging is caused by a sudden and substantial decrease in available NADPH, which means that cells cannot maintain a stable oxidation state. If borne out by further work, these findings could have a significant impact on how we think about the aging process, and could require researchers to rethink how they study aging.

lifespan of yeast, we will obtain a spatial and temporal read-out of the prevailing oxidation conditions during the aging process. We should also be able to uncover protein targets whose oxidative thiol modifications might contribute to the physiological alterations that are observed in aging organisms and might even be able to establish a clear correlation between onset and extent of oxidative stress and lifespan.

The chronological lifespan of *S. cerevisiae* represents a popular model system for analyzing aging in postmitotic cells. Chronological lifespan is defined as the length of time that non-dividing cells remain viable in a high metabolic state (*Fabrizio and Longo, 2007*; *Fontana et al., 2010*). In support of the free radical theory of aging, chronological lifespan decreases in yeast strains lacking the oxidant scavenging enzymes superoxide dismutase (SOD) or catalase (*Longo et al., 1996*) and increases when glutathione or SOD levels are elevated (*Harris et al., 2003*). Also, caloric restriction, a nearly universal measure to extend lifespan, has been shown to significantly increase chronological lifespan in yeast (*Fontana et al., 2010*). Although the molecular mechanism by which caloric restriction extends lifespan has not been elucidated, one unifying trait among calorically restricted organisms is a significantly increased oxidative stress resistance (*Barja, 2002*).

In this study, we used chronologically aging *S. cerevisiae* to determine the onset, extent, and targets of protein oxidation in postmitotic aging cells. By monitoring the thiol oxidation status of almost 300 different protein thiols, we discovered that yeast cells undergo a global redox collapse that

leads to massive thiol oxidation in almost 80% of identified proteins several days prior to cell death. Cluster analysis revealed that this general protein oxidation is preceded by the oxidation of a subset of conserved proteins, one of which is NADPH-dependent thioredoxin reductase, a key enzyme in maintaining redox homeostasis. Redox metabolite and NADPH studies suggested that protein oxidation is triggered by a decrease in cellular NADPH concentration. Consistent with this hypothesis, caloric restriction delayed NADPH decrease, early protein oxidation, global redox collapse, and cell death. Our results suggest that the decrease in cellular NADPH levels initiates oxidation of the cellular redox machinery and triggers system-wide oxidation events, which significantly precede cell death.

## Results

### Using OxICAT to monitor the in vivo redox status of proteins during the chronological lifespan of yeast

Chronological lifespan measurements of *S. cerevisiae* wild-type and mutant strains suggested that ROS might affect and potentially even determine the postmitotic lifespan of yeast (*Longo et al., 1997*; *Fabrizio and Longo, 2007*). We therefore decided to apply the quantitative redox proteomic technique OxICAT to monitor the redox status of our previously identified yeast protein thiols during the chronological lifespan of this organism. OxICAT is based on the differential modification of in vivo reduced and in vivo oxidized cysteine thiols, respectively with isotopically light $^{12}C$ and isotopically heavy $^{13}C$ versions of the isotope-coded affinity tag (ICAT) thiol alkylating reagent (for scheme see *Figure 1—figure supplement 1A*). This differential thiol trapping with ICAT is followed by a tryptic digest of the proteins contained in the cell lysate and the purification of all ICAT-labeled peptides using an affinity tag. Liquid chromatography combined with mass spectrometry (MS) and MS/MS analysis is used to separate and identify the ICAT-labeled peptides, and to quantify the ratio of in vivo reduced to oxidized protein thiols in individual peptides. Because this ratio is unaffected by changes in relative protein amounts, OxICAT is uniquely suited to simultaneously monitor changes in the thiol oxidation status of hundreds of proteins over time.

We had previously identified and quantified the steady-state oxidation level of almost 400 thiol-containing peptides in about 290 different yeast proteins localized to various cellular and subcellular compartments (*Brandes et al., 2011*). We now reasoned that by simultaneously monitoring the thiol oxidation status of all these proteins during postmitotic aging, we should be able to track potential redox changes and identify affected proteins and pathways, provided that oxidant levels and redox conditions changed significantly during the lifespan of the organism. Moreover, by cultivating yeast cells under different conditions, including conditions such as caloric restriction, which has previously been shown to alter chronological lifespan (*Fontana et al., 2010*), we should be able to reveal any correlation between onset and extent of oxidative stress and the lifespan of yeast.

We therefore cultivated the wild-type yeast strain DB746 under three different conditions: standard, caloric restriction, or with a water 'starvation' diet. Under standard conditions, 2% glucose SCD media is used and DB746 cells maintain their high metabolic, postdiauxic state until they die (mean lifespan ~7 days) (*Fabrizio and Longo, 2007*) (*Figure 1A*). Under caloric restriction (CR), 0.5% glucose SCD media is used, which increases respiration, promotes higher oxidative stress resistance, and extends lifespan (mean lifespan ~11 days) (*Figure 1A*) (*Fabrizio and Longo, 2003*). With the starvation diet, cells are provided 2% glucose SCD media for 2 days followed by incubation in water. Under these conditions, yeast cells switch to a hypometabolic state (stationary phase) and show dramatically increased lifespan (mean lifespan > 15 days) (*Figure 1A*) (*Fabrizio et al., 2003*). We monitored growth for the first 24 hr (*Figure 1—figure supplement 1B*), and took samples for our OxICAT analysis at 24-hr intervals starting during exponential growth (day 0) and continuing until 10–20% of cells were dead (day 4 in standard media, day 7 in caloric restriction media) or up to day 10 in water.

To initially determine whether and when ROS levels change during yeast chronological aging, we analyzed the thiol oxidation status of glyceraldehyde-3-P dehydrogenase (GapDH, TDH) as this protein has some of the most redox-sensitive cysteines in yeast and other organisms (*Brandes et al., 2011*). GapDH contains two redox-sensitive cysteines (the active site Cys150 and the nearby Cys154), and both of these are found in the same tryptic peptide (GapDH[144–160]). We previously suggested that these cysteines form an intramolecular disulfide bond during peroxide stress in vivo (*Brandes et al., 2011*). During the first 2 days of cultivation under either standard or caloric restriction conditions, we found that the GapDH[144–160] peptide was predominantly labeled with two light ICAT molecule and less

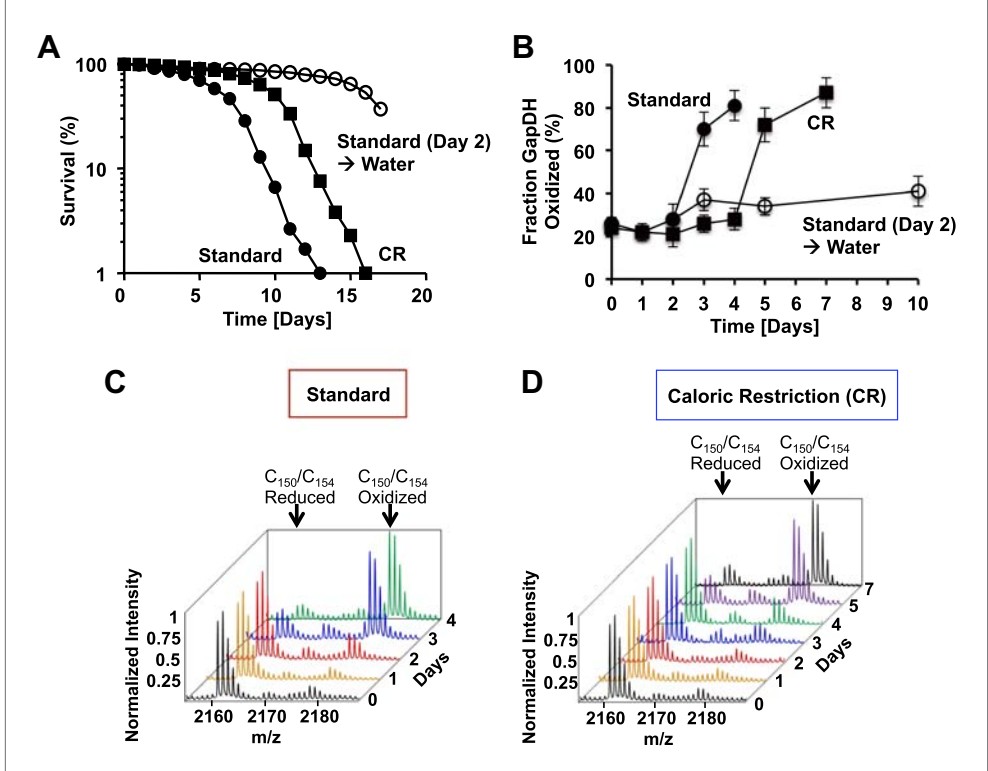

**Figure 1**. The active site cysteines of GAPDH become increasingly oxidized during the chronological lifespan of yeast. Chronological lifespan of *S. cerevisiae* strain DBY746 was monitored under either 2% glucose standard conditions (closed circles) or 0.5% glucose caloric restriction (CR) conditions (closed squares). Alternatively, cells were cultivated under 2% glucose standard conditions for two days, washed and resuspended in water to induce hypometabolic cultivation conditions (open circles). Cell aliquots were taken at the indicated time points and (**A**) viability was determined using propidium iodide (PI) staining or (**B**–**D**) the thiol oxidation status of Cys150/Cys154 in GapDH was quantified by differential thiol trapping using OxICAT. Representative MS spectra of the differentially ICAT-labeled GapDH[144–160] peptide containing Cys150 and Cys154 are shown in panels C and D. The mass peak at *m/z* 2161.13 corresponds to the reduced GapDH[144–160] peptide in which both cysteines are labeled with light ICAT. The 18 Da heavier mass peak at *m/z* 2179.13 corresponds to the oxidized GapDH[144–160] peptide in which both cysteines are labeled with heavy ICAT.

The following source data and figure supplements are available for figure 1:

**Source data 1.** Average oxidation status with standard deviation of protein thiols identified in at least three biological replicates under each cultivation condition.

**Figure supplement 1**. (**A**) Schematic overview of the OxICAT procedure.

than 30% of the two cysteines were calculated to be oxidized (**Figures 1B–D**). However, within the next 24 hr (i.e., day 3) of cultivation under standard conditions, about 70% of the GapDH peptide was labeled with two heavy ICAT molecules, indicating that both cysteines were oxidized (**Figure 1C**, compare red and blue trace). This extent of oxidation is very similar to that observed in yeast cells treated with 0.5 mM $H_2O_2$ for 15 min (**Brandes et al., 2011**). On day 4 under standard conditions, over 80% of all GapDH molecules were oxidized, indicating that by that time, glycolysis is dramatically reduced. In contrast, the oxidation status of GapDH from yeast cells cultivated under caloric restriction conditions remained low for the first 4 days of cultivation (**Figures 1B,D**). Then, however, significant oxidation occurred (also within a 24-hr time window), with over 70% of all GapDH molecules affected by day 5 and almost 90% affected by day 7 (**Figure 1D**, purple trace and black trace, respectively). Yeast cells cultivated in standard media for 2 days and then switched into water showed no significant increase in GapDH oxidation over the time span that was monitored by OxICAT (**Figure 1B**, open

circles and *Figure 1—Source data 1*), suggesting that under hypometabolic conditions, cells maintain GapDH in its reduced and active state over extended periods of time.

## Oxidation of the thiol redox proteome: a global and early event in the chronological lifespan of yeast

To determine whether the observed thiol oxidation is restricted to a subset of particularly redox-sensitive proteins or affects a wider range of yeast proteins, we analyzed the redox status of all of our previously identified protein thiols during the chronological lifespan. We reproducibly identified 286 of these protein thiols in all samples taken under standard 2% glucose conditions (i.e., all four replicates at five different time points) (*Figure 1—Source data 1*). Most of these protein thiols were also identified in the four replicates and seven time points taken from cultures cultivated under caloric restriction conditions (i.e., 263 peptides), and 100 of these peptides were reproducibly identified under hypometabolic water starvation conditions as well (*Figure 1—Source data 1*). We discovered that the majority of our identified protein thiols found in standard and caloric restriction conditions followed an oxidation pattern similar to that of GapDH's active site cysteines (*Figure 2A*). The protein thiols were largely reduced (*Figure 2A*, blue) for the first 48 hr post log phase (i.e., day 0) under standard conditions or for the first 96 hr post log phase under caloric restriction conditions, and then became suddenly oxidized within a 24-hr period (*Figure 2A*, red). After one more day of cultivation, the majority of these protein thiols were then oxidized to over 80% (*Figure 2A* and *Figure 1—Source data 1*). Shifting cells to water at day 2 of cultivation in 2% glucose SCD media prevented this sudden onset of oxidation, and proteins showed only a minor increase in their thiol oxidation state, which persisted until at least day 10 (*Figure 2A* and *Figure 1—Source data 1*).

## Cellular processes affected by the redox collapse

Based on the large number of proteins that are affected by the apparent redox collapse in yeast cells, it is not surprising that many of the identified proteins are involved in central physiological processes. For instance, we found over 40 oxidation-sensitive proteins that play crucial roles in protein translation, including 27 different thiol-containing 40S, 60S, and 54S ribosomal proteins, several translation initiation (e.g., TIF11, GCD1) and elongation factors (e.g., EFT2, TEF1, TEF3), translational activators (e.g., GCN1), and numerous tRNA synthetases (e.g., MES1, KRS1, VAS1). Oxidation of these proteins most likely affects the rate and extent of protein synthesis in chronologically aging yeast cells. Moreover, we found countless metabolic enzymes, including ACO1, PGK1, IDP1, PDC1, and TPI1 to become oxidized, likely affecting processes ranging from the Krebs cycle and the pentose phosphate pathway to fatty acid and amino acid synthesis. Also, numerous oxidation-sensitive proteins that we identified are known to be involved in maintaining protein homeostasis, including chaperones (e.g., YDJ1, HSP78, SSA1/2, SSB1/2, SSE1/2, SSZ1), prolyl isomerases (e.g., FPR1/2, CPR1, 6), components of the proteasome complex (e.g., PRE10, PUP2), and ubiquitination machinery (e.g., UBC4), or serve as part of the cellular antioxidant response (e.g., PRDX, thioredoxin reductase1/2) (*Figure 1—Source data 1*). Many of these proteins have previously been found to contain redox-sensitive cysteines (*Lindahl et al., 2011*). In fact, of the 290 different protein thiols that we monitored in our study, over 33% have been confirmed to be redox-sensitive in yeast or other organisms (*Figure 1—Source data 1*). Moreover, an additional 20% of our identified cysteines are localized to proteins that have been found to contain redox-sensitive cysteines but whose redox-sensitive cysteines have not yet been fully identified. These percentages were obtained by comparing our list of aging-oxidized protein thiols with the recently published RedoxDB database, which compiled and manually curated over 2100 proteins with over 2300 redox-sensitive cysteines from different eukaryotic organisms (*Sun et al., 2012*). The high degree of overlap between our identifications and the list of previously identified redox-sensitive proteins in eukaryotes makes us confident about the specificity of our method. Our finding that over 80% of an unbiased population of thiol-containing yeast proteins have the capacity to become significantly oxidized under physiologically relevant growth conditions (*Figure 1—Source data 1*) suggests that reversible cysteine oxidation, such as that detected with our OxICAT method, is a much more widespread event than previously anticipated.

Despite the limited supply of nutrients in the stationary phase, oxygen consumption measurements indicate that chronologically aging yeast cells are metabolically active in this phase (*Fabrizio et al., 2003*; *Fabrizio and Longo, 2003*). Consistent with earlier ATP measurements conducted in chronologically aging yeast cells (*Goldberg et al., 2009*), we found that intracellular ATP is indeed maintained

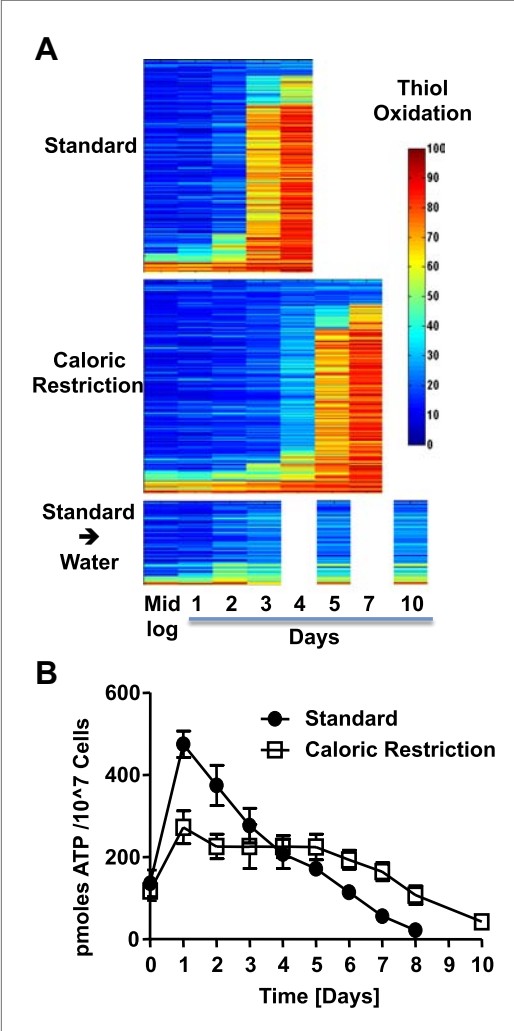

**Figure 2**. The redox homeostasis collapses early in postmitotic yeast. (**A**) DBY746 cells were grown with initial glucose concentrations of either 2% (standard) or 0.5% (caloric restriction). At defined time points, samples were taken for OxICAT analysis (see **Figure 1**, legend). To determine the thiol oxidation status of cells under hypometabolic conditions, cells were cultivated in standard media for 2 days, washed, then shifted to water prior to taking samples for OxICAT analysis. Each identified peptide is depicted as a bar colored according to its in vivo oxidation state from 0% (blue) to 100% (red) (**Figure 1—Source data 1**). Peptides are organized by their oxidation pattern in standard conditions. The color presentation was done by Matlab. (**B**) Cells were cultivated under standard (filled circles) or caloric restriction (open squares) conditions. Cell aliquots were taken at the indicated time points and total cellular ATP levels were determined as described in 'Material and methods'.

at levels equivalent to or above those present during exponential growth for up to 5 days when cultivated under standard postmitotic growth conditions and for at least 7 days under caloric restriction conditions (**Figure 2B**). These results demonstrate that chronologically aging yeast cells are not starving and are able to maintain their energy resources and survive for extended periods of times despite (or possibly even because of) a heavily oxidized proteome.

## Protein cluster analysis reveals distinct waves of protein oxidation

Analysis of the kinetics of protein thiol oxidation revealed that the majority of protein thiols in yeast follow the trend observed for GapDH oxidation: the thiol oxidation state is low for the initial 2 days of cultivation then suddenly increases by day 3 in standard media or by day 5 in caloric restriction media. In addition, however, we noticed several protein thiols whose oxidation appeared to precede this global wave of oxidation by 24–48 hr (**Figure 2A**). To investigate the significance of this finding in detail, we clustered all identified protein thiols according to their oxidation kinetics. This cluster analysis is based on a *k*-means with Euclidean distance algorithm (**Saeed et al., 2006**). More than 95% of our identified protein thiols clustered into one of seven distinct oxidation clusters (named A–G) and most protein thiols maintained their cluster assignment independent of the cultivation condition (standard or caloric restriction media) (**Figure 3** and **Figure 1— Source data 1**). Clusters A–C contained the majority of our identified peptides and included all those thiol groups that revealed a sudden onset of oxidation by either day 3 in standard media or day 5 in caloric restriction media. The peptides only differed in their extent of oxidation within the first 24 hr after onset of oxidation (cluster A: >50% oxidation; cluster B: <50% oxidation) or in their initial oxidation level (clusters A and B: <20% oxidation; cluster C: >40% oxidation). About 10% of identified peptides preceded this general oxidation trend by 24–48 hr (**Figure 3**, clusters D and E and **Table 1**). These peptides showed significantly higher oxidation levels at either day 1 (**Figure 3**, cluster D) or day 2 (**Figure 3**, cluster E) compared to their oxidation states at day 0. Importantly, most of these protein thiols that were subject to early oxidation under one condition were found to be early oxidation targets under the other cultivation condition as well. These peptides are of particular interest because their oxidation may not just serve as an early warning of age-induced changes in the oxidative status of cells, but might induce changes in their activity, which are involved in controlling or triggering the oxidation of proteins in general. Only about 15% of the identified cysteine-containing

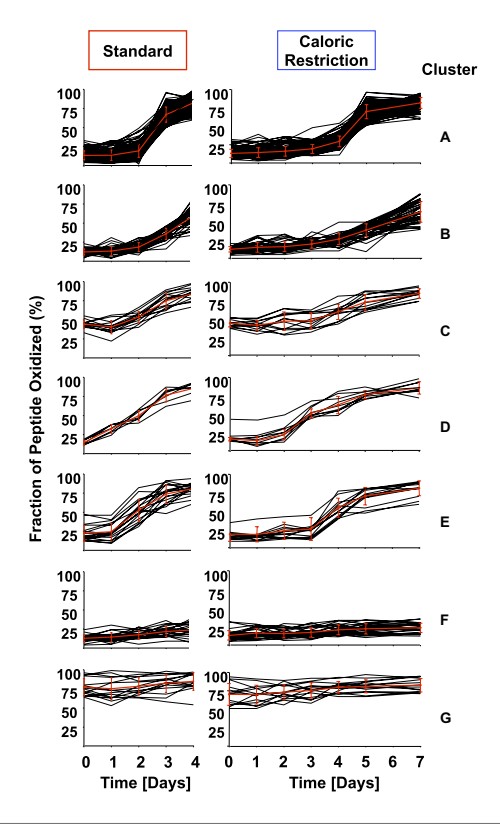

**Figure 3**. Cluster analysis of identified peptides reveals early oxidation targets. All identified peptides in cultures cultivated under standard or calorically restricted conditions were clustered using the *k*-means (Euclidean distance) clustering algorithm. Each peptide is displayed by a black line; the red line represents the average of the cluster. Over 70% of peptides fall into clusters A–C. Cluster A: all peptides with less than 30% thiol oxidation during log phase and an increase in oxidation to more than 50% on day 3 (day 5 under caloric restriction conditions) of cultivation; Cluster B: all peptides with less than 30% thiol oxidation during log phase and an increase in oxidation more than 50% on day 4 (day 6 under caloric restriction conditions) of cultivation. Cluster C: all peptides with ~50% thiol oxidation during log phase and a significant increase in oxidation on day 3 (day 5 under caloric restriction conditions) of cultivation. Cluster D: all peptides that show an at least 1.5-fold increase in thiol oxidation beginning on day 1 (day 3 under caloric restriction conditions) of cultivation. Cluster E: all peptides that show an at least 1.5-fold increase in thiol oxidation beginning on day 2 (day 4 under caloric restriction conditions) of cultivation. Peptides in Clusters F and G remain reduced or oxidized, respectively. The majority of peptides identified under standard or calorically restricted conditions fall into the same clusters (see *Figure 1—Source data 1* for details). Peptides in cluster D or E are listed in *Table 1*.

yeast peptides remained reduced (*Figure 3*, cluster F) or stayed oxidized (*Figure 3*, cluster G) throughout the course of the incubation (*Figure 1—Source data 1*).

Analysis of the subcellular distribution of the proteins in the individual clusters did not reveal any significant correlation between the location of the proteins and their oxidation pattern (apart from the expected accumulation of proteins of the endoplasmic reticulum and secreted proteins in cluster G) and reflected well the general subcellular distribution of all identified proteins. To begin to understand what specifies the cluster behavior of the individual protein thiols, we performed a bio-informatic analysis of the identified cysteine thiols. In-depth analysis of the cysteine's $pK_a$-values or the localization of the respective cysteine residues within the proteins (e.g., surface exposed vs buried) was unfortunately hampered by the limited number of available protein structures (*Brandes et al., 2011*). We thus performed a sequence analysis of the 10 amino acids surrounding each identified thiol group. To increase our sample size, we combined clusters with similar oxidation trends and initial oxidation levels (e.g., cluster A with cluster B, cluster D with cluster E). The most noticeable difference among the thiol groups in the individual clusters was that those cysteine thiols that became oxidized early (clusters D and E) as well as protein thiols with high steady-state levels of oxidation during exponential growth (clusters C and G) had a significant ($p<0.05$) accumulation of additional cysteines in close proximity (*Figure 4A, B*). In contrast, those protein thiols that stayed reduced (cluster F) or were initially reduced and followed the general oxidation trend (clusters A and B) almost completely lacked the presence of nearby additional cysteines. Chi-square-analysis confirmed the significance of this finding and excluded the dominance of any other amino acid type apart from cysteine in close vicinity of the identified thiol (*Table 2A,B*). These results suggest that the early oxidation of a subset of yeast proteins is likely triggered by a more oxidizing redox environment, which causes those protein thiols that can undergo stabilizing disulfide bonds with nearby cysteines to accumulate in their oxidized state. One alternative explanation, that a sudden surge in ROS, such as peroxide, causes oxidation of a group of particularly ROS-sensitive proteins, appears less likely as only a few of our previously identified peroxide-sensitive cysteines and none of our most peroxide-sensitive peptides (i.e., GapDH) were found among the group of early oxidation targets (*Table 1*). It also remains to be seen what makes the about 8% protein thiols in cluster F so resistant to protein

**Table 1.** Early oxidation targets in yeast

| Gene (Cys) | Protein | Loc. | 2% glucose (standard) | | | | | 0.5% glucose (CR) | | | | | | |
|---|---|---|---|---|---|---|---|---|---|---|---|---|---|---|
| | | | D0 | D1 | D2 | D3 | D4 | D0 | D1 | D2 | D3 | D4 | D5 | D7 |
| CCT4 (399) | T-complex protein 1 subunit delta | C | 12 | 24 | 58 | 87 | 86 | 12 | 14 | 13 | 11 | 39 | 72 | 82 |
| ARO2 (221)* | Chorismate synthase | C | 13 | 13 | 31 | 71 | 74 | | | | nd | | | |
| CDC48 (115) | Cell division control protein 48 | ER, C | 12 | 29 | 55 | 75 | 93 | 13 | 15 | 14 | 28 | 62 | 68 | 90 |
| CCT8 (336) | T-complex protein 1 subunit theta | C | 47 | 40 | 77 | 76 | 89 | | | | nd | | | |
| TRR (142;145) | Thioredoxin reductase | C/M | 33 | 33 | 65 | 77 | 82 | 34 | 39 | 42 | 44 | 73 | 80 | 86 |
| UBC4 (108) | Ubiquitin-conjugating enzyme E2 4 | N | 19 | 23 | 48 | 70 | 81 | 20 | 15 | 22 | 24 | 49 | 73 | 88 |
| YCR090C (124) | UPF0587 protein | C, N | 34 | 39 | 67 | 76 | 88 | | | | nd | | | |
| LYS2 (614) | L-aminoadipate-semialdehyde DH | C | 15 | 22 | 49 | 79 | 86 | 14 | 11 | 12 | 48 | 56 | 74 | 81 |
| YDJ1 (185;188)* | Homologue of DnaJ | C | 46 | 46 | 82 | 89 | 75 | 45 | 44 | 51 | 70 | 73 | 83 | 88 |
| MES1 (353) | Methionyl-tRNA synthetase | C | 23 | 34 | 64 | 86 | 82 | 20 | 21 | 31 | 61 | 78 | 80 | 98 |
| OLA1 (43) | Uncharacterized GTP-binding protein | C | 17 | 12 | 46 | 91 | 83 | 20 | 10 | 22 | 29 | 53 | 78 | 88 |
| PAA1 (51;55) | Polyamine N-acetyltransferase 1 | C | 26 | 36 | 53 | 48 | 61 | 18 | 15 | 25 | 52 | 61 | 71 | 83 |
| PRB1 (36) | Cerevisin | V | 18 | 23 | 57 | 91 | 94 | 12 | 14 | 26 | 23 | 77 | 80 | 88 |
| PUT2 (162) | δ-1-pyrroline-5-carboxylate DH | M | 12 | 10 | 35 | 60 | 66 | | | | nd | | | |
| RPL10 (49) | 60S protein L10 | C | 15 | 15 | 31 | 73 | 89 | 21 | 17 | 28 | 23 | 59 | 80 | 90 |
| RPL42B (74)* | 60S protein L42 | C | 15 | 22 | 49 | 86 | 86 | 15 | 19 | 25 | 49 | 54 | 77 | 93 |
| RPS11B (58) | 40S protein S11 | C | 18 | 17 | 37 | 77 | 81 | 14 | 17 | 27 | 25 | 36 | 83 | 86 |
| RPS22B (72) | 40S protein S22-B | C | 13 | 9 | 34 | 62 | 72 | 32 | 29 | 36 | 32 | 41 | 72 | 79 |
| SES1 (413;414)* | Seryl-tRNA synthetase | C | 22 | 23 | 55 | 66 | 79 | 20 | 19 | 17 | 43 | 65 | 71 | 81 |
| HEM1 (386) | 5-aminolevulinate synthase | M | 13 | 21 | 58 | 77 | 88 | | | | nd | | | |
| IDP1 (398) | Isocitrate dehydrogenase 1 | M | 18 | 19 | 41 | 61 | 85 | 16 | 18 | 32 | 25 | 49 | 81 | 80 |
| KGD1 (983) | 2-oxoglutarate dehydrogenase E1 | M | 25 | 21 | 49 | 57 | 84 | 18 | 18 | 19 | 16 | 57 | 54 | 70 |
| FAS2 (917) | Fatty acid synthase subunit alpha | C, M | 22 | 16 | 30 | 62 | 86 | 14 | 8 | 15 | 59 | 81 | 87 | 72 |
| ERG13 (300) | Hydroxymethylglutaryl-CoA synthase | ER | 17 | 38 | 47 | 80 | 91 | 16 | 18 | 30 | 44 | 55 | 70 | 84 |
| FUS2 (371) | Nuclear fusion protein FUS2 | N | 18 | 33 | 57 | 68 | 80 | 9 | 18 | 12 | 20 | 31 | 64 | 91 |
| LAP4 (202) | Vacuolar aminopeptidase 1 | V | 19 | 38 | 45 | 83 | 86 | 25 | 35 | 27 | 22 | 39 | 61 | 81 |
| PYC2 (218) | Pyruvate carboxylase 2 | C | 11 | 28 | 46 | 81 | 93 | 13 | 11 | 22 | 47 | 66 | 77 | 81 |
| TEF1 (409)* | Elongation factor 1-alpha | C | 13 | 38 | 40 | 62 | 70 | 18 | 30 | 17 | 26 | 36 | 57 | 68 |
| GapDH(150;154)† | Glyceraldehyde-3-P Dehydrogenase | C | 26 | 22 | 28 | 70 | 81 | 24 | 22 | 21 | 26 | 28 | 72 | 87 |

*Peroxide sensitive (**Brandes et al. 2011**).

†Follows the general oxidation pattern.

All cluster D and E proteins thiols whose oxidation kinetics significantly preceded the general oxidation trend are listed. Thiol oxidation states, which are at least 2-fold higher as compared to day 0 or at least 1.5 fold higher as compared to day 0 and exceeding a total oxidation of 60% are shaded. Standard deviations can be found in **Figure 1—Source data 1**.

oxidation, as it is conceivable that some of these proteins might play a role in promoting longevity. Analysis of the nature of these proteins, which are listed in *Figure 1—Source data 1*, did not reveal any striking trends in regards to their functions or subcellular localizations. We assume that their oxidation resistance is based on potentially unusual structural features influencing the reactivity of the cysteine thiol, such as an abnormal $pK_a$-value or a very buried nature. However, at this point, we lack sufficient structural information on this group of proteins to draw any firm conclusions why some of their cysteines are so oxidation resistant.

## Thioredoxin reductase: an early oxidation target in yeast

Oxidation of at least 28 proteins significantly preceded the general oxidation of proteins under standard or caloric restriction conditions (*Figure 3*, clusters D and E and *Table 1*). Of these

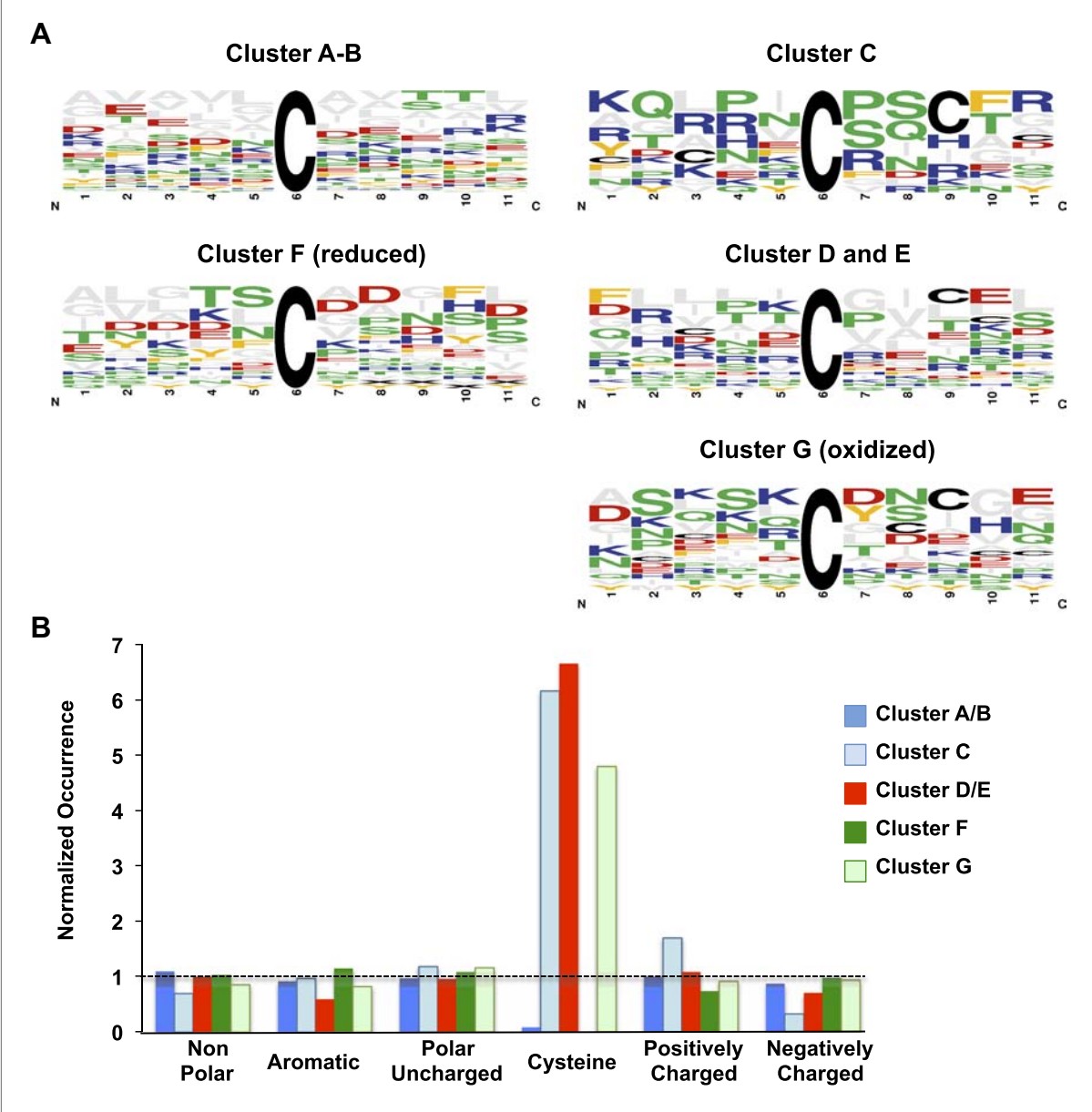

**Figure 4**. Comparison of sequence conservation between individual protein clusters. Analysis of sequence conservation (**A**) and amino acid type (**B**) in sequence fragments spanning five amino acids up- and downstream of the thiol group whose oxidation status was determined by OxICAT. Peptide sequences from clusters A and B were combined as were sequences from clusters D and E. (A) Sequence logos of the 11-amino acid peptides were aligned at the position of the identified cysteine. The color code corresponds to the amino acid type, with Cys shown in black, negatively charged amino acids shown in red, positively charged amino acids shown in blue, non-polar amino acids shown in grey, aromatic amino acids shown in yellow, and polar amino acids shown in green. The residue order in each column corresponds to the relative occurrence of the residue in the specific position. The height of the amino acid corresponds to its relative frequency at the specific position. The logos were created using WebLogo (***Crooks et al., 2004***). (B) The relative amino acid occurrence, excluding the OxICAT-identified cysteine, in the sequence fragments was analyzed. As in (A), the amino acids were grouped according to their characteristics and the occurrence of the amino acid type was normalized to the distribution of the same amino acid type in the entire library of sequence fragments. A value of 1 indicates that the occurrence of a specific amino acid is identical to the occurrence of this group of amino acids in the total sequence library.

early-oxidized proteins, 20 had oxidation states of more than 45% at day 2 of cultivation, which was 1.5- to 3.8-fold higher than their oxidation status during exponential growth. One of these early oxidation targets is the highly conserved enzyme thioredoxin reductase, the key component of the thioredoxin system. Although we cannot exclude that oxidation of any one of the other early oxidation targets

**Table 2.** Chi-square analysis of amino acid type distribution in sequence fragments containing the identified thiol group according to clusters

| Clusters | A–B | C | D–E | F | G |
|---|---|---|---|---|---|
| Table 2A | | | | | |
| A–B | 1 | | | | |
| C | **2.5E-19** | 1 | | | |
| D–E | **3.4E-28** | **0.0227** | 1 | | |
| F | 0.4871 | **4.7E-07** | **0.0001** | 1 | |
| G | **1.9E-18** | **0.0300** | 0.4358 | **0.0085** | 1 |
| Table 2B | | | | | |
| A–B | 1 | | | | |
| C | **6.0E-05** | 1 | | | |
| D–E | 0.5066 | **0.0113** | 1 | | |
| F | 0.3781 | **0.0003** | 0.1248 | 1 | |
| G | 0.2757 | **0.0169** | 0.4025 | 0.6238 | 1 |

Table 2A. Amino acid distribution was analyzed in the sequence fragments spanning five amino acids up- and downstream of the cysteine thiol (**Figure 4**), whose oxidation status was determined by OxICAT. The identified cysteine thiol was not included in the analysis. Table 2B. Chi-square analysis of the amino acid type distribution in the same sequence fragments analyzed in **Table 2A** removing any cysteines from our analysis. p-values obtained from the chi-square analysis of distribution of different amino acid types, positively and negatively charged, polar, non-polar, aromatic amino acids and cysteines (for **Table 2A** only) in clusters A through G (**Figure 3**). Degrees of freedom are 5 (**Table 2A**) and 4 (**Table 2B**), respectively. Significantly different distributions are shown in bold (p<0.05, a = 0.95).

directly or indirectly affects or even controls *S. cerevisiae* lifespan, we decided to focus our subsequent studies on thioredoxin reductase, as this enzyme is the central player in maintaining cellular redox homeostasis. Loss of thioredoxin reductase activity has been shown to cause widespread protein oxidation (**Holmgren and Lu, 2010**). We found that the oxidation of thioredoxin reductase's two active site cysteines, which are arranged in the prototypical C-X-X-C motif, raised sharply by about twofold to 65% at least 24 hr before the general redox collapse began (**Figure 5**, compare blue and red trace) and was close to 80% at day 3 of cultivation under standard conditions (**Figure 5** and **Table 1**). The same trend was observed under caloric restriction: although oxidation of thioredoxin reductase was delayed by 48 hr relative to what was seen under standard conditions, it again preceded the general redox collapse by about 24 hr (**Table 1**). Note that a shift to extreme caloric restriction (water) partially reversed the early oxidation of thioredoxin reductase that was noticeable at day 2 of cultivation (**Figure 5**). Within 24 hr upon shift into water, we observed a significant reduction in thioredoxin reductase's oxidation status, which reached levels that were only slightly above the initial oxidation levels of thioredoxin reductase in exponentially growing yeast cells (**Figure 5**). These low levels of oxidation were then maintained for the remainder of the experiment, consistent with our previous observations that yeast proteins do not become significantly oxidized upon shift into water during the time period tested. These results suggest that early interventions restore the activity of thioredoxin reductase and also prevent the collapse of cellular redox homeostasis.

To further elucidate what role if any thioredoxin reductase plays in postmitotic lifespan, we decided to generate yeast mutants lacking either the cytosolic (thioredoxin reductase 1) or the mitochondrial (thioredoxin reductase 2) form of thioredoxin reductase in our DBY746 strain. We found that while Δ*trr2* mutants grew like wild-type yeast cells and had a wild-type like chronological lifespan, deletion of the cytosolic TRR1 homologue (Δ*trr1*) caused severe growth defects and a significantly shortened lifespan (**Figure 5—figure supplement 1**). Moreover, we observed the generation of healthy-looking Δ*trr1* suppressors with very high frequency. In fact, cultivation of Δ*trr1* deletion mutants without the potential generation of suppressors was only possible when the medium was supplemented with cysteine, presumably to maintain proteins in their reduced state. This cysteine requirement, however, made the investigation of the oxidation status of proteins in Δ*trr1* strains very difficult, and the interpretation of the effects of a *ttr1* deletion on lifespan very challenging. Organisms lacking

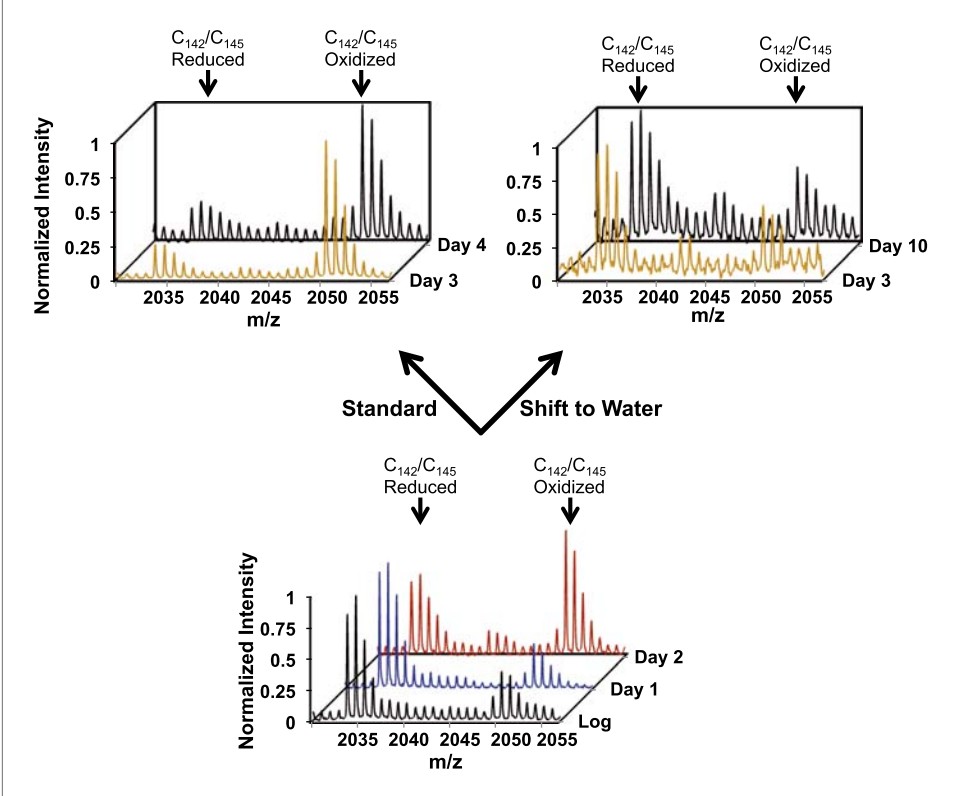

**Figure 5**. Early oxidation of thioredoxin reductase is reversible in vivo. Yeast strain DBY746 was cultivated under standard conditions for 2 days (lower panel). Then, the culture was split and either continued to be cultivated in standard media (upper left panel) or shifted to water (upper right panel) to induce hypometabolic cultivation conditions. Representative MS spectra of the differentially ICAT-labeled thioredoxin reductase peptides containing the two active site cysteines Cy142/Cys145 before and after the shift are shown. Within 24 hr after shifting cultures to hypometabolic cultivation conditions (day 3), the increased thiol oxidation of thioredoxin reductase's active site cysteines observed at day 2 is largely reversed.

The following figure supplements are available for figure 5:

**Figure supplement 1**. Role of thioredoxin reductase in the chronological lifespan of yeast.

thioredoxin reductase have previously been shown to be either severely compromised in growth or non-viable (*Holmgren and Lu, 2010*). This illustrates the essential role that maintenance of the cellular redox homeostasis plays in organisms and explains why directly investigating the role of thioredoxin reductase in the lifespan has remained great challenge.

## Depletion of cellular NADPH levels: the trigger for early protein oxidation events?

The NADPH-dependent thioredoxin system is one of two highly conserved multi-enzyme systems that contribute to the maintenance of cellular redox homeostasis in most pro- and eukaryotic organisms. The other system consists of NADPH-dependent glutathione reductase, several small glutaredoxins, and the cysteine-containing tripeptide glutathione (GSH), which together with its oxidized counterpart GSSG determines the cellular redox potential (*Martensson and Meister, 1989*; *Merad-Boudia et al., 1998*). To investigate potential changes in the redox potential of postmitotic yeast cells cultivated in standard or caloric restriction media, we took samples during chronological aging of DBY746 yeast cells and determined total GSH and GSSG concentrations at the same time points at which we previously analyzed the thiol oxidation states. As shown in *Figure 6A*, we found that during exponential growth, the glutathione redox potential did not differ between yeast cells cultivated in standard or caloric restriction conditions. However, by day 1 of cultivation in standard conditions, yeast cells

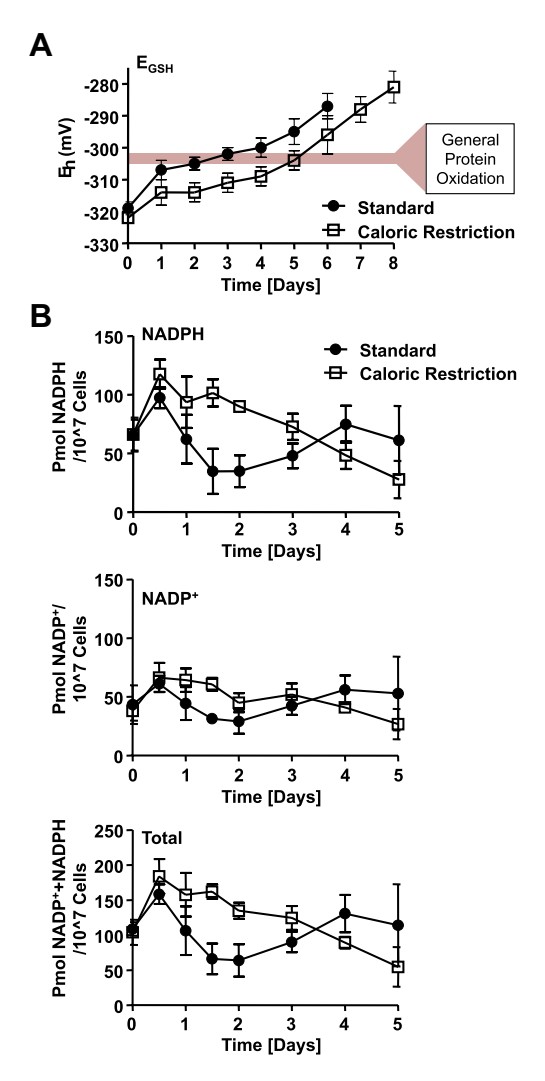

**Figure 6**. Loss of cellular NADPH might trigger redox collapse. Strain DBY746 was cultivated under standard (full circles) or caloric restriction (open squares) conditions as described in *Figure 1*. At the time points indicated, samples were taken for (**A**) whole cell analysis of GSH and GSSG levels or (**B**) NADPH/NADP⁺ measurements. The glutathione redox potential $E_{GSH}$ was calculated using the Nernst equation. Data points are the average of at least three independent experiments: bars indicate standard deviation.

showed an at least 15 mV increase in their overall redox potential, which induces a significant shift in the thiol/disulfide or thiol/sulfenic acid equilibrium of proteins that are in equilibrium with the GSH/GSSG couple. In contrast, yeast cells cultivated under caloric restriction conditions experienced a much smaller initial increase in the cellular redox potential. This result agrees with previous studies that revealed a more oxidizing environment for cells in standard media as compared to caloric restriction media (*Magherini et al., 2009*). Importantly, cells cultivated in caloric restriction media showed a 48-hr delay to reach the same redox potential observed in cells cultivated in standard media (*Figure 6A*). This delay is consistent with our previously observed 48-hr delay in protein oxidation in yeast cells cultivated in caloric restriction media as opposed to standard media.

Both thioredoxin reductase and glutathione reductase draw their reducing power from NADPH, making the oxidation status of these systems ultimately dependent on cellular NADPH and NADP⁺ levels. We therefore measured the levels of NADPH and NADP⁺ in yeast samples cultivated under both standard and caloric restriction conditions. We found that exponentially growing yeast cells have very similar levels of NADPH and NADP⁺ independent of their initial glucose availability, and that these levels increased over the next 12 hr under both cultivation conditions (*Figure 6B*). In this time frame, yeast cells undergo a diauxic shift from glucose-driven fermentation to ethanol-driven respiration. Cells induce NADH kinases such as UTR1 and POS5, glucose-6-phosphate dehydrogenase (ZWF1), and the cytosolic NADPH-dependent isocitrate dehydrogenase (Idp2) to increase NADPH production and regeneration (*Gasch et al., 2000*). In cells cultivated under standard conditions, the intracellular levels of NADPH then rapidly decreased over the next 24-hr time period. In contrast, in cells cultivated under caloric restriction conditions, NADPH levels decreased with much slower rates and reached concentra-

tions comparable to those observed in standard media with about a 48-hr delay (*Figure 6B*). This drop in intracellular NADPH levels coincided well with the initial oxidation of thioredoxin reductase and the alteration in the cellular redox potential. It is unclear why yeast cells grown under standard conditions show increased levels of both NADP⁺ and NADPH at day 4 of cultivation in standard media (*Figure 6B*). It is conceivable that cells cannibalize and hence may take up metabolites from surrounding dying cells. Alternatively, oxidation and potential inactivation of NADPH-utilizing enzymes might serve as negative feedback loop and lead to the observed increase in NADPH levels. In summary, these results suggest that early changes in cellular NADPH levels might serve as trigger for the initial oxidation of thioredoxin reductase and changes in the cellular redox potential, which subsequently leads to the

redox collapse observed in postmitotic yeast cells. Cultivation under caloric restriction conditions appears to delay the decrease in cellular NADPH levels, delays the redox collapse of the yeast proteome, and extends lifespan.

## Discussion

In this study, we used a quantitative redox proteomic approach combined with metabolic measurements to assess a time line of physiological redox events that occur in aging non-dividing cells, using chronologically aging *S. cerevisiae* as a model system. We made the very surprising observation that early during the chronological aging process and significantly before cell death sets in, yeast cells undergo an abrupt loss in redox homeostasis as indicated by the massive oxidation of the large majority of thiol-containing cytosolic, nuclear, and mitochondrial proteins. Importantly, this oxidation event is significantly delayed by caloric restriction and even more so by a shift to hypometabolic cultivation conditions, suggesting that maintenance of redox homeostasis might contribute to the lifespan extending effects of these regimens. To begin to understand what physiological event(s) trigger the observed redox collapse, we compared the kinetics of oxidation in almost 300 different protein thiols. We noted that thiol oxidation occurs in at least two waves. The first wave, which affects only a very small subset of identified thiol-containing proteins (<10%), hit cells cultivated in standard media within 24–48 hr after reaching exponential growth. At this point, cells had stopped dividing (*Figure 1—figure supplement 1B*), transitioned to respiratory growth, and NADPH levels, which transiently increased during the diauxic shift, had started to decrease significantly (*Figure 6B*). The second wave of oxidation, which occurred about 24 hr later, affected nearly 70% of the remaining identified yeast protein thiols (*Figure 2*). Yet as observed before, yeast cells were able to maintain their energy resources (i.e., ATP levels) and to survive for several more days after the collapse. Bioinformatic analysis revealed that many of the very early oxidation targets are cysteines that form part of a $C-X_{2/3}-C$ motif (e.g., thioredoxin reductase, CCT4, CCT8, YdJ1, RPL42, PAA1, MES1). This cysteine motif is often found in disulfide oxidoreductases, redox-sensitive transcription factors, and many metal binding proteins, and confers considerable redox sensitivity to proteins (*Sanchez et al., 2008*). Hence, this cysteine motif allows many of these proteins to form transient disulfide bonds within the otherwise reducing environment of the cytosol. Based on the popular free radical theory of aging, we first suspected that a sudden surge in or accumulation of peroxide might be the cause of the early oxidation of these proteins. However, we found that only 5 of the 28 early oxidation targets were previously identified to contain peroxide-sensitive thiols (*Table 1*, indicated with asterisk). Moreover, protein thiols that we had previously identified to be highly peroxide-sensitive, such as the active site cysteines of GapDH or AHP1 (a thiol-peroxidase that undergoes reversible disulfide bond formation upon peroxide detoxification) were not among the early oxidation targets in yeast (*Figure 1—Source data 1*). These results suggested that elevated peroxide production is unlikely the cause of the early oxidation event. Instead, we noted that the early oxidation event is significantly preceded by a loss in cellular NADPH, the electron donor of the NADPH-dependent thioredoxin system. Moreover, we found that under conditions of caloric restriction, each of the individual processes was time-delayed by about 48 hr: NADPH decrease, early protein (i.e., thioredoxin reductase) oxidation, and the collapse of the thiol redox proteome. These findings raised the intriguing possibility that these processes are directly connected (*Figure 7*) and that loss of NADPH might be the trigger for the observed redox collapse. Consistent with this idea, analysis of the cellular GSH/GSSG ratio in chronologically aging yeast cells, which is dependent on the NADPH-dependent glutathione reductase, revealed a pro-oxidizing shift in the GSH redox potential that coincided with the decrease in cellular NADPH levels. As before, this pro-oxidizing shift was significantly delayed in calorically restricted growth conditions (*Figure 7*). These findings would explain how a reduction in caloric intake, which transiently increases cellular NADPH levels (*Figure 6B*), is able to extend maintenance of the cellular redox balance and might contribute to lifespan extension. It is noteworthy that similar pro-oxidizing shifts in thioredoxin and glutathione systems have also been observed in aging rodents (*Cho et al., 2003*; *Rohrbach et al., 2006*; *Rebrin and Sohal, 2008*), and might, at least in part, be explained by the observed decrease in cellular NADPH levels in aging rats (*Parihar et al., 2008*). Consistent with our studies, caloric restriction at least partially reversed the detected changes in redox status, shifted the glutathione pool to a more reducing redox potential relative to cultures grown in 2% glucose, and increased cellular NADPH levels (*Someya et al., 2010*). These results suggest that the observed decrease in cellular NADPH

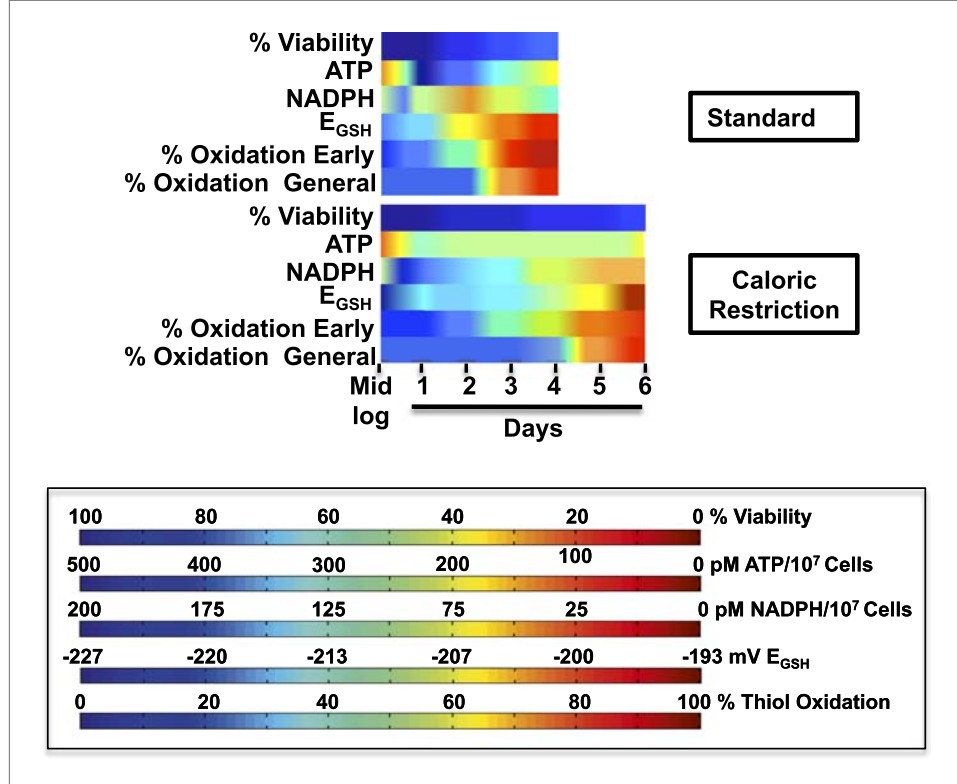

**Figure 7**. Timeline of redox events in chronologically aging yeast cells. The graphs shown provide a comparative assessment of cell viability (based on data shown in *Figure 1A*), ATP levels (*Figure 2B*), NADPH levels (*Figure 6B*), $E_{GSH}$ measurements (*Figure 6A*), and thiol oxidation states of representative early (i.e., PYC2) and general targets (e.g., GapDH/TDH) (*Table 1*) during the chronological lifespan of yeast strain DBY746 under standard and caloric restriction conditions. A colored scale for each assessed parameter is provided.

levels and the concomitant increase in cellular redox potential is not just a yeast-specific event but might be shared with other aging organisms as their metabolism changes. The fact that yeast cells undergo the very same pattern of NADPH decrease, Trr1 oxidation and redox collapse (albeit time-delayed) when cultivated in 2% glucose or in 0.5% glucose medium also argues against the possibility that medium acidification, which has been discussed to accelerate chronological aging in yeast, is a contributing factor to the observed effects, as medium acidification is severe in 2% glucose yet insignificant in 0.5% glucose (*Burtner et al., 2009*).

It was intriguing to observe that early protein oxidation is, at least in its initial stage, a fully reversible event in yeast. Moreover, we found that more than 80% of viable cells were recovered from day 3- and day 4-old cultures despite an almost fully oxidized thiol proteome. It has been suggested that oxidative thiol modifications, such as those detected by our OxICAT method, might serve a beneficial purpose for cells by preventing irreversible thiol modifications that would ultimately lead to protein degradation and potentially cell death (*Gallogly and Mieyal, 2007*). It is thus tempting to speculate that the initially reversible thiol modifications that we observe in chronologically aging yeast cells might in fact represent a 'pro-active' response of yeast cells to protect their proteins against irreversible protein modifications and damage, and thereby extend lifespan. Quantitative studies conducted in diamide-treated HeLa and HEK cells revealed that over 50% of protein thiols undergo reversible thiol modifications in response to diamide stress, providing evidence that the combined redox proteome of mammalian cells has a higher redox buffering capacity than glutathione (*Hansen et al., 2009*). While the authors were unable to conclude whether this high redox buffering capacity of protein thiols is the work of a few high abundance, highly cysteine-enriched proteins or a contribution of the majority of thiol-containing proteins, our data suggest that the majority of yeast protein thiols have the capacity to undergo reversible redox modifications and thereby serve as redox buffer. These results suggest

that chronological aging might represent a physiological stress condition that utilizes and stresses the redox buffering capacity of the thiol proteome. At this point, we also cannot exclude that oxidation of some key proteins might actually contribute to the potentially beneficial down-regulation of cellular processes that would otherwise negatively affect yeast lifespan and shorten it even more. One such example would be methionyl-tRNA synthetase (MES1), an enzyme involved in translational initiation (*Delarue, 1995*) whose oxidation of Cys353 reaches over 65% at day 2 of cultivation in standard conditions (*Table 1*). Intriguingly, Cys353 is part of a zinc finger-like $CX_2C$-$X_9$-$CX_2C$ motif whose substitution in the *E. coli* homologue MetRS causes a strong decrease in enzyme activity (*Fourmy et al., 1993*). Down-regulation of protein translation has been shown to contribute to increased oxidative stress resistance and to extend lifespan in replicative aging yeast and other organisms (*Steffen et al., 2008*).

In summary, our results suggest that early changes in cellular NADPH levels might serve as a trigger for the initial thioredoxin reductase oxidation, which subsequently leads to the redox collapse observed in postmitotic yeast cells. Cultivation of yeast cells under caloric restriction conditions appears to delay the decrease in cellular NADPH levels and hence delays the redox collapse of the yeast proteome. At this point, it is unclear which event(s) cause the initial drop in intracellular NADPH levels that trigger the redox collapse. Moreover, it remains to be determined whether this is a controlled pro-survival response that extends an otherwise even shorter lifespan, or the first step on the final path to destruction.

## Material and methods

### Strains, cell growth and chronological lifespan measurements

*S. cerevisiae* strain *EG103* (DBY746; *MATα, leu2-3 112 his3Δ1 trp1-289a ura3-52*) was cultivated in synthetic complete dextrose (standard SCD) medium, which consists of 0.67% yeast nitrogen base supplemented with complete amino acid mix (*Guthrie, 2002*) and 2% wt/vol glucose at 30°C. To cultivate yeast under caloric restriction conditions, glucose concentration was decreased to 0.5% wt/vol. Chronological lifespan was monitored as previously described (*Fabrizio and Longo, 2007*). Cell aliquots were taken each day and viability was assessed using propidium iodide (PI) staining (*Deere et al., 1998*). Viability is given as the percent of cells that are unstained by PI over the total number of cells in the optic field. Deletion mutants of thioredoxin reductase 1 and 2 (TRR1 and TRR2) were constructed in EG103 by using homologous recombination of a PCR product containing the ClonNAt resistance marker (*Goldstein and McCusker, 1999*).

### Differential thiol trapping of proteins during chronological lifespan and OxICAT analysis

EG103 cells were grown in standard or caloric restriction medium at 30°C with continuous shaking. Once cells reached mid-logarithmic phase ($OD_{600}$ of 0.5), the first cell aliquot was harvested (corresponding to day 0). All further cell aliquots were harvested in 24-hr intervals (day 1, 2, etc.). For each aliquot, $5 \times 10^7$ cells (total volume adjusted for changes in cell density) were harvested directly onto 10% (wt/vol) trichloroacetic acid (TCA) to stop all thiol-disulfide exchange reactions. TCA-precipitated samples were incubated on ice for 30 min and the OxICAT thiol trapping protocol including mass spectrometry and data analysis was conducted as described previously (*Brandes et al., 2011*).

### Cluster analysis

The open-source software TIGR MultiExperimentViewer v4.4 (MEV; http://www.tm4.org/mev/) (*Saeed et al., 2006*) and the algorithm *k*-means clustering with Euclidean distance (implemented in MEV) were used for clustering analysis of peptides listed in *Figure 1—Source data 1*. Values missing in those lists as a result of insufficient MS quantification were predicted with Coupled Two Way Clustering (CTWC) (Weizmann Institute of Science, Israel; http://ctwc.weizmann.ac.il/) (*Getz et al., 2000*). The prediction was based on five neighbors when more than 30% of the values were known.

### Analysis of intracellular ATP concentrations

Intracellular ATP levels were determined as previously described (*Bondar and Mead, 1974*; *Yang et al., 2002*). Results are expressed as mean ± standard deviation of three independent experiments.

### Determination of intracellular glutathione concentrations

For determination of intracellular GSH and GSSG concentrations, $10 \times 10^7$ cells were harvested at the indicated time points and the metabolites were measured after derivatization with iodoacetic acid and

dinitrofluorobenzene followed by HPLC analysis (*Garg et al., 2010*). Redox potentials were calculated using the Nernst equation: $E_h = E_0 + RT/2F \ln[GSSG/(GSH)^2]$ with $E_0 = -240$ mV for the GSH/GSSG couple.

## Analysis of intracellular NADP(H) levels

Samples containing $1 \times 10^7$ cells were harvested by low speed centrifugation from chronologically aging yeast cultures, washed with cold phosphate buffered saline (PBS), and resuspended to a final $OD_{600}$ of 10 using the extraction and lysis buffer provided by the fluorescent NADPH/NADP detection kit from Cell Technology Inc. (Mountain View, CA). Cells were lysed with glass beads. Extraction and detection of $NADP^+/NADPH$ was conducted according to the manufacturer's protocol.

## Acknowledgements

We thank Dr. James Bardwell for critically reading the manuscript. LC-MS/MS and MS analysis was performed by the Michigan Proteome Consortium (www.proteomeconsortium.org), which was supported in part by funds from the Michigan Life Sciences Corridor.

## Additional information

### Funding

| Funder | Grant reference number | Author |
|---|---|---|
| National Institutes of Health | AG027349 | Ursula Jakob |
| National Institutes of Health | HL58984 | Ruma Banerjee |
| National Institutes of Health | AG000114 | Heather Tienson |
| Human Frontier Science Program | | Dana Reichmann |
| European Molecular Biology Organization | | Dana Reichmann |

The funders had no role in study design, data collection and interpretation, or the decision to submit the work for publication.

### Author contributions

NB, Acquisition of data, Analysis and interpretation of data, Drafting or revising the article; HT, Acquisition of data, Analysis and interpretation of data, Drafting or revising the article; AL, Acquisition of data; VV, Acquisition of data; DR, Analysis and interpretation of data, Drafting or revising the article; RB, Analysis and interpretation of data; UJ, Conception and design, Analysis and interpretation of data, Drafting or revising the article

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
