## [Author Response]

*1. The authors argue that it is not possible to lower the activity of NADPH thioredoxin reductase (Trr) as loss of this enzyme affects fitness in their strain background. However, it should be possible to lower the dose of Trr, e.g. ,using weaker promoters, or in heterozygous diploids to test a potential role of this enzyme in aging. Alternatively, the authors could consider the overexpression of the non-phosphorylated NADP-dependent glyceraldehyde-3-phosphate dehydrogenase, gapN, from either S. mutans or B. cereus, although there is some concern that the results of overexpression experiments might be difficult to interpret. At the very least the authors should expand their discussion on the limitations of their study*.

We thank the reviewers for these suggestions. We completely agree that directly analyzing the role of cytosolic thioredoxin reductase 1 (TRR1) in chronological lifespan would be desirable and we have tried extensively in the past to address this question.

As part of this work, we have conducted lifespan measurements of TRR1 null mutants (see Figure 5–figure supplement 1 in the revised version). The data in Figure 5–figure supplement 1 clearly show that absence of cytosolic Trr1 causes a significantly shortened chronological lifespan. This result, at a minimum, illustrates that maintaining redox balance is an important contributor to overall fitness. But how do we distinguish between lack of overall fitness of TRR1 mutants and accelerated aging? Distinguishing between fitness and aging effects of mutants is a challenge that the whole field struggles with. It is one of the chief reasons that mutants that prolong lifespan are easier to interpret than those that shorten lifespan. In the specific case of TRR1, there is the added problem that at this point it is unclear how many of the cells developed suppressor mutations during the long-term cultivation in the absence of cysteine. Many others in the field have faced the exact same dilemma, and, as far as we know, nobody has been able to directly test the influence of thioredoxin reductase on lifespan. These considerations make us hesitant to publish the lifespan data. Instead, to address the reviewer's comment, we have now added the statement below to our revised manuscript:

“To further elucidate what role if any thioredoxin reductase plays in postmitotic lifespan, we decided to generate yeast mutants lacking either the cytosolic (thioredoxin reductase1) or the mitochondrial (thioredoxin reductase2) form of thioredoxin reductase in our DBY746 strain. […]This illustrates the essential role that maintenance of the cellular redox homeostasis plays in organisms and explains why directly investigating the role of thioredoxin reductase in the lifespan has remained a great challenge.”

We have also begun to attempt to alter cellular NADPH levels and test their influence on lifespan. However, these are very challenging, time-intensive experiments, and, as the reviewers point out, interpretation of the data will likely be very difficult. If they are successful, they will form the basis of a future manuscript.

*2. The authors should include a discussion of the nature of the proteins present in cluster F. Proteins in this cluster are remarkably resistant to the general redox status of the cell and hence might play an important role in promoting longevity*.

We agree that the fact that the cluster F protein thiols are very oxidation resistant is quite remarkable. We assume that this oxidation resistance is based on some potentially unusual structural features (e.g., a very high pKa, or a buried nature), which we cannot decipher at this point because we do not have enough structural information on this group of proteins to make a statistically convincing case. Some of the cysteines are located in proteins that also have highly oxidation-sensitive cysteines, and we have too little information about which of these cysteines play a functionally or structurally more dominant role. We have now briefly discussed these points in our revised manuscript: “To begin to understand what specifies the cluster behavior of the individual protein thiols […].”

*3. A previous study by Burtner and colleagues (2009) has argued that it is acetate toxicity and pH change, rather than oxidation, that causes a loss of viability in post-mitotic cells. Therefore, it seems warranted that the authors discuss the different models. Experimentally, the authors could consider testing if ras2Δ and sch9Δ mutants that extend stationary phase survival also delays the drop in NADPH and oxidation. In this context, deletions of SPT5 would also be very informative as this mutant displays both reduced NADPH and acetate levels. The Burtner model would predict an extended life span whereas the current work would predict the opposite*.

The reviewers point out correctly that yeast cells accumulate acetate in the medium during the fermentative phase of yeast growth, which they subsequently use as carbon source after switching to oxidative phosphorylation. The accompanying pH drop in the medium of yeast cells cultivated in 2% standard glucose has been discussed to affect the chronological lifespan of yeast. However, the precise reason(s) for this observation are still unclear and the role of acetic acid in chronological lifespan remains highly controversial (see recent review by Longo, Shadel, Kaeberlein and Kennedy, *Cell Metabolism*, 2012). As we do not argue that oxidation causes the loss in viability but rather discuss the possibility that early oxidation might serve as a more pro-active response to down-regulate cellular processes and, hence, prolong an otherwise potentially even shorter lifespan, we do not see the need for getting involved in this discussion. We do want to add, however (and have now done so in the revised manuscript) that according to our experiments, yeast undergo the very same pattern of NADPH decrease, Trr1 oxidation, redox collapse, when cultivated in 2% glucose or 0.5% glucose medium. This finding clearly argues against the possibility that medium acidification is involved in these processes as cultivation of yeast cells in 2% glucose has been shown to decrease the pH of the cultivation medium while cultivation of cells in 0.5% CR medium does not appear to affect the pH of the medium (Burtner et al., 2009). We have addressed this subject in our revised manuscript:

“The fact that yeast cells undergo the very same pattern of NADPH decrease, Trr1 oxidation and redox collapse (albeit time-delayed) when cultivated in 2% glucose or in 0.5% glucose medium also argues against the possibility that medium acidification, which has been discussed to accelerate chronological aging in yeast, is a contributing factor to the observed effects, as medium acidification is severe in 2% glucose yet insignificant in 0.5% glucose (Burtner et al., 2009).”

The reviewers also suggested to test a variety of different mutants, i.e., *ras2Δ* and *sch9Δ* mutants, as well as to investigate how deletion of Stb5p (we assume that this is the protein that the reviewer meant rather than SPT5), an oxidation-sensitive zinc cluster-containing transcription factor, which up-regulates the pentose phosphate pathway and hence NADPH levels, will affect chronological lifespan and the oxidation status of yeast proteins. We agree that an obvious extension of this work is to now figure out what causes the NADPH levels to change and to elucidate whether this is a planned, pro- active response, or a natural consequence of metabolism. However, these studies are beyond the scope of this study and the subject of ongoing work in our lab. In our manuscript, we have provided the first evidence that changes in cellular NADPH levels set into motion a cascade of events that lead to a cellular redox collapse, and that these events can be significantly delayed under conditions that extend lifespan.